# Algebraical Entropy and Arrow of Time

**DOI:** 10.3390/e24111522

**Published:** 2022-10-25

**Authors:** Merab Gogberashvili

**Affiliations:** 1Department of Physics, Javakhishvili Tbilisi State University, 3 Chavchavadze Avenue, 0179 Tbilisi, Georgia; gogber@gmail.com; 2Andronikashvili Institute of Physics, 6 Tamarashvili Street, 0177 Tbilisi, Georgia

**Keywords:** normed algebras, non-associativity, entropy, time arrow, 02.10.Ud, 03.65.Fd, 03.67.-a, 65.40.Gr

## Abstract

Usually, it is supposed that irreversibility of time appears only in macrophysics. Here, we attempt to introduce the microphysical arrow of time assuming that at a fundamental level nature could be non-associative. Obtaining numerical results of a measurement, which requires at least three ingredients: object, device and observer, in the non-associative case depends on ordering of operations and is ambiguous. We show that use of octonions as a fundamental algebra, in any measurement, leads to generation of unavoidable 18.6 bit relative entropy of the probability density functions of the active and passive transformations, which correspond to the groups G2 and SO(7), respectively. This algebraical entropy can be used to determine the arrow of time, analogically as thermodynamic entropy does.

## 1. Introduction

The basic microscopic physical laws are time reversible [1], while the second law of thermodynamics is able to describe macroscopic irreversible processes through the change of entropy. It seems that the thermodynamic explanation of the time asymmetry is implausible [2]; if we repeat an experiment sufficiently many times, we might be able to record statistically rare events where the state of the system reverts to its initial conditions. We need some fundamentally time-asymmetric theories. Perhaps the future quantum gravity would be able to explain the irreversibility [3], since quantum theory incorporates a fundamental arrow of time in the measurement process. Recent technological advances, such as quantum weak measurements [4], enable us to explore quantum measurements in a time-symmetric picture. Experimentally, the absolute irreversibility for measurements performed on a quantum many-body system was demonstrated [5].

Observers are powerful players in quantum theory, since a quantum system picks a specific state only after observation that breaks superpositions. There exist so-called epistemic interpretations of quantum theory, which treat our world just as expressions of information [6]. If one allows for an indeterminate influence of observation on the observed phenomenon, the concept of an objective world existing independently of observation becomes questionable [7]. The phenomenon of wave reduction associated with observation on quantum systems introduces fundamental irreversibility in the observed behavior.

In this paper, we consider the possibility to link the microscopic entropy and the arrow of time with the non-associativity of nature at a fundamental level. A measurement process means manipulation of a physical system with some device by an observer, i.e., requires at least three operators that are corresponding to these ingredients. The final result of a measurement is some real number obtained from multi-component functions (e.g., the norm of a wave function), i.e., any physical observable depends also on the algebra we use.

Note that for our consideration it is not crucial to introduce the observer dependent irreversibility, non-associativity could be manifested in any complicated or strongly interacted system containing more than two particles (e.g., in some regimes of quantum chromodynamics, quantum gravity, and so on). Any particle (an excitation of quantum field), before a measurement by interactions with another particle, can be considered as entangled with the rest of the universe (multi-particle system). So, any measurement should be described by the product of at least three wave functions, and we presuppose that some non-associative quantum field theory can exist.

## 2. Algebraical Entropy

To account for ambiguities of non-associative products in some non-associative quantum field theory, we want to introduce the concept of algebraical entropy. Ordinary thermodynamic entropy reflects the number of accessible microstates of the system in study, while algebraical entropy can account for the number of measurements needed to obtain the values of the accessible parameters of non-associative functions.

Considered in this paper, algebraical entropy (corresponding to non-associativity) should not be confused with the algebraic entropy introduced for discrete dynamical systems [8,9]. If a dynamical system is sufficiently complex, any initial imbalance is eliminated due to energy dissipation, which is closely associated with the concept of thermodynamic irreversibility. In this case, the algebraic entropy intends to measure disorders created by the discrete dynamical system and is a global index of complexity for the evolution map.

Let us estimate the algebraical entropy of non-associativity on the example of the algebra of octonions [10,11,12,13,14], which is the interesting mathematical structure for physical applications [15,16,17,18]. Normed division algebras can be regarded as the universal mathematical structures in physics, since they provide a natural framework to describe geometry, physical variables and their relationships [19,20,21,22,23,24,25,26].

It is known that, in addition to the usual algebra of real numbers, there are three other unique normed division algebras: complex numbers, quaternions and octonions. The real numbers are ordered, commutative and associative, but for each next mentioned algebra, one such property is lost. The element *X* of any normed real algebra can be written as the linear combination of the basis elements with the real coefficients, $X^{0}$ and $X^{n}$:(1)$$X=X^{0}e_{0}+\sum_{n}X^{n}e_{n}\;,$$
where the unit basis element, $e_{0}=1$, is real and the other $e_{n}$ are hyper-complex. The dimensions of the vector part of complex numbers, quaternions and octonions are n= 1, 3 and 7, respectively. The square of the unit element in (Equation 1) is always positive, while the squares of all hyper-complex basis units mainly are considered to be negative (similar to the ordinary complex unit *i*),
(2)$$e_{0}^{2}=1\;,\;e_{n}^{2}=-e_{0}=-1\;.$$ In normed algebras, the conjugation operation is defined, which does not affect unit element but changes the sign of all hyper-complex basis units,
(3)$$e_{0}^{*}=e_{0}\;,\;e_{n}^{*}=-e_{n}\;.$$ Then, using the multiplication and conjugation rules of $e_{n}$, the positively defined (Euclidean) quadratic form (norm) can be defined,
(4)$$N=XX^{*}=X_{0}^{2}+\sum_{n}X_{n}^{2}\;.$$ Any element of the division algebra (Equation 1) might be expressed in the polar form:(5)X=Nα(θ),
where
(6)$$\alpha \left(\theta \right)=e^{\epsilon \theta }\;,$$
is the element with the unit norm, $N_{\alpha }=1$. In (Equation 6), the hyper-complex element ϵ ($\epsilon^{2}=-1$) represents the unit *n*-vector. The unit exponential elements (Equation 6) can be used to define rotations [14,19,20],
(7)$$X^{\prime }=\alpha \left(\frac{\theta^{d}}{2}\right)X\;\alpha^{*}\left(\frac{\theta^{d}}{2}\right)\;,$$
by the real angles $\theta^{d}$ (*d* is the number of independent generators), which are defined to lie in the interval (−π,π]. Note that, in general, the mappings (Equation 7) are not elementary; that is, they can rotate more than one pair of basis elements.

If the norm (Equation 4) of *X* is given, the conditional probability that the random phase angles in (Equation 6) (which vary in the interval [0,2π]) obtain the specific values (e.g., all $\theta^{d}$ are zero and the real part of (Equation 1), $X_{0}$, is known) equals to
(8)$$P\left(\theta^{d}|N\right)=\frac{1}{{\left(2\pi \right)}^{d}}\;.$$ So, the continuous random angles $\theta^{d}$ have the uniform distributions (probability density functions)
(9)$$f\left(x|N\right)=\frac{1}{{\left(2\pi \right)}^{d}}\;,\;\;\left(\int_{S_{d}}dx^{d}\;f\left(x|N\right)=1\right)$$
where $S_{d}$ is the support set of f(x|N). Then, for normed algebras we can define continuous analogue of Shannon’s discrete entropy—the conditional differential entropy [27],
(10)$$H\left(\theta^{d}|N\right)=-\int_{S_{d}}dx^{d}\;f\left(x|N\right)\log_{2}f\left(x|N\right)=\log_{2}\left(2\pi \right)\;d\approx 2.7\;d\;\mathrm{bit}\;,$$
which in a single measurement roughly captures the amount of ’lack of information’ on an observable that corresponds to the norm of a multi-component number *X*, i.e., on the values of all parameters of *X*. For example, for the case of complex functions, we have just one phase parameter (d=1) and, according to (Equation 10), the numerical result of a single measurement exhibits U(1)-ambiguity in complex plain and generates 2.7 bit of differential entropy.

In quantum physics, one must be very careful about the meaning of ’lack of information’, since one cannot speak about the precise state of a system before making measurements. So, the entropy should be viewed as a measure of the amount of information that we can in principle extract from the system [28]. Using of differential entropy (Equation 10) for this purpose is problematic, e.g., $H(\theta^{d}|N)$ is inconstant under change of variables; we need some measure of information that is not dependent on the used coordinate system. One such quantity is the Kullback–Leibler divergence, also called relative entropy of the probability distributions f(x) and g(x), which is defined as [27]:(11)$$D\left(f||g\right)=\int_{S}dx\;f\left(x\right)\log_{2}\frac{f(x)}{g(x)}\;,$$
where *S* is the support set of f(x). The relative entropy (Equation 11) is a measure of how the probability distribution f(x) is different from the reference probability distribution g(x).

## 3. Arrow of Time

We suppose that algebraical entropy that corresponds to non-associativity (Equation 11) can be linked with time asymmetry [29]. Geometry is not independent from observations, and any change in information is stored as change in an emerging metric [30,31,32,33]. Allowing operators and wave functions of a physical theory on non-associative algebra, expressions of expectation values (products of some three quantities: $\psi_{1}$, $\psi_{2}$ and $\psi_{3}$) become ambiguous. Not being able to determine expectation value of an operator makes it inherently unobservable, and the measurement outcome cannot be predicted [34,35]; we need to introduce some brackets rule, since, in general,
(12)$$\left(\psi_{1}\psi_{2}\right)\psi_{3}\neq \psi_{1}\left(\psi_{2}\psi_{3}\right)\;.$$ Moreover, in the expressions of the Feynman amplitudes *M* for strongly interacting systems, which contain products of several field operators $\psi_{n}$, we need a rule for ordering the products. For example, we can assume that the brackets are arranged in accordance with the left rule [36]:(13)$$M_{L}∼\psi_{1}\psi_{2}\psi_{3}...\psi_{n}∼\left(...(\psi_{1}(\psi_{2}(\psi_{3}...\right)\psi_{n}\;,$$
i.e., all opening brackets are in the extreme left-hand position. In such a model, the Feynman amplitude calculated by the left rule (Equation 13) will be different from the right amplitude,
(14)$$M_{R}∼\psi_{1}\left(\psi_{2}\left(\psi_{3}...\psi_{n}\right)...\right)\;.$$ This property is similar to the postulation of time ordering, i.e. forward and backward probabilities of a particle reaction are not equal [29]. Non-associative parts in Hamiltonians may be interpreted as not preserving probabilities over time.

As we already mentioned in the introduction, in any measurement process the three systems: an object, memory (or apparatus) and the observer are involved, i.e., on an elementary level, to obtain numerical a result, we need to calculate products of at list three functions (Equation 12), which in general is ambiguous. On the other hand, expressions containing any combination of only two functions are always associative. This can explain, at least within the epistemic quantum models [6], why the basic equations of two-particle reactions might be time-reversal, while a measurement that introduces an additional operator always leads to the arrow of time.

To explore fundamental algebraical uncertainties arising in measurements, which are independent of the reference parameters of an observer, we can compare the distribution f(x|N) of the angles of the active rotations (Equation 9) (generated by the left and right products on the elements of algebra (Equation 7) and forming the automorphism group), with the reference distribution g(x|N) that corresponds to the passive transformations of components of (Equation 1) (coordinate transformations that preserve the norm (Equation 4)),
(15)$$X_{0}\to X_{0}^{\prime }\;,\;X_{n}\to X_{n}^{\prime }\;.\;\;\left(X_{0}^{{}^{\prime }2}+\sum_{n}X_{n}^{{}^{\prime }2}=X_{0}^{2}+\sum_{n}X_{n}^{2}\right)$$ For normed algebras, the tensorial transformations (Equation 15) form the Lie groups SO(n). The number of generators of SO(n), i.e,. the number of independent angles $\theta^{d}$ in (Equation 9), equals to
(16)$$d=\frac{1}{2}n\left(n-1\right)\;.$$ Recall that n= 1, 3 and 7 for complex numbers, quaternions and octonions, respectively.

For complex numbers and quaternions (spinors), both sets of the transformations, (Equation 7) and (Equation 15), are equivalent and form the groups SO(2) (or U(1)) and SO(3), respectively. We should note that for complex numbers U(1) operates as effective automorphisms, beyond identity and complex conjugation [37,38]. Then, for the complex and quaternionic functions f(x|N)=g(x|N) and the relative entropies (Equation 11) for them are zero. This means that, since norms and all rotation angles are known, by continuous measurements we are able to obtain all values of the parameters $X_{0}$ and $X_{n}$ in (Equation 1), i.e., no algebraical uncertainty arising due to the use of the multi-dimensional numbers *X* is left.

For octonions (non-associative bi-spinors), to represent the active rotations (Equation 7), which preserve the multiplicative structure of the algebra, we would need the transformations to be automorphisms [19,21]. Due to non-associativity, not all tensorial transformations of the parameters (Equation 15) (that form the Lie group SO(7)—the rotation group in 7 dimensions with the dimension 21) represent real rotations, only the transformations that have a realization as associative multiplications should be considered [39]. Automorphisms of octonions form the subgroup of SO(7), the Cartan’s smallest exceptional Lie group $G_{2}$ with the dimension 14 [40]. According to (Equation 16), the number of independent generators of SO(7) is 21, while the number of generators of $G_{2}$ is
(17)d=14. So, for octonions, active and passive transformations of a general element *X* are not equal, and the algebraical entropy can be calculated using the parameters of $SO\left(7\right)/G_{2}$. For the probability density functions of the active and passive transformations, (Equation 7) and (Equation 15), we can use the constant distributions:(18)$$f_{G_{2}}\left(x|N\right)=\frac{1}{{\left(2\pi \right)}^{14}}\;,\;\;g_{SO(7)}\left(x|N\right)=\frac{1}{{\left(2\pi \right)}^{21}}\;.$$ The relative entropy (Equation 11) of the probability distributions (Equation 18), where the dimension of the support set is calculated by (Equation 17), has the value:(19)$$D\left(f||g\right)=\int_{d}dx^{d}\;f_{G_{2}}\left(x\right)\log_{2}\frac{f_{G_{2}}\left(x\right)}{g_{SO(7)}\left(x\right)}=\left(21-14\right)\log_{2}\left(2\pi \right)\approx 18.6\;\mathrm{bit}\;.$$ We see that if we use octonionic functions, obtaining numerical results (norms of octonionic signals) in any measurement generates ≈18.6 bit algebraical entropy, which corresponds to the information that is principally inaccessible. So, the non-associativity introduces unavoidable algebraical entropy and in principle can be used to determine the arrow of time, analogically as thermodynamic entropy does.

## 4. Conclusions

To conclude, in this paper we have considered the possibility that at a fundamental level, nature is not only non-commutative, but non-associative as well. In any measurement, one needs to evaluate products of at least three quantum operators, which symbolically correspond to the object under consideration, measuring device and the observer. In general, the result of the product of three non-associative elements is ambiguous. The necessity of some ordering rule for products can be interpreted as the introduction of a time arrow; left and right products, corresponding to direct and reverse processes are different. At the same time, any product that contains only two elements is well defined. This explains why basic physical laws for two-particle interactions can be symmetric in time, while the complex systems exhibit irreversibility. To account for this property, we have introduced the notion of algebraical entropy, arising due to using the real norms of multi-dimensional numbers for physical observables. For octonions, we found that any measurement generates unavoidable ≈18.6 bit algebraical entropy that corresponds to the principally inaccessible information carried by the non-associative functions.

At the end, note that, while in general the phase of a complex function cannot be uniquely predicted from its magnitude, in some cases, to reduce the amount of experimentally obtained information causality and linearity conditions are introduced. This implies the condition of analyticity, which results in Kramers–Kronig or Bode-type dispersion relations between the real and imaginary parts [41] that can reduce the algebraical entropy in (Equation 11). However, it is problematic to define analyticity conditions for quaternions [42] and especially for octonions. So, in this paper we do not want to consider the issues of causality and dispersion relations.

## Data Availability

Not applicable.

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
