# Peer review of "Algebraical Entropy and Arrow of Time"

_entropy, 2022, doi:10.3390/e24111522_

Round 1
Reviewer 1 Report (Previous Reviewer 2)
I thank the author for the response to my previous comments and revisions to the manuscript.
I still feel that the manuscript lacks the connection to a feasible measurement process that could be implemented in a lab in order to measure the entropy generated as proposed in the manuscript. I understand there could be several difficulties around proposing such an experiment given that most of our present experimental avenues do not use octonion algebra to describe physical processes. Even then I find it helpful that the author makes a connection to scattering processes involving more than two-particle interactions as a potential avenue to experimentally probe the entropy associated to the non-associativity of the algebra of octonians. Therefore if possible, I would like to invite the author to further expand on potential experimental signatures for this result. If such a proposal is infeasible at the moment, I believe it would be helpful for the readers if the author wishes to comment on the difficulties around proposing such an experiment.
Author Response
Of course, it will be interesting to propose a decisive experiment that will prove our model. But, it is difficult to manage an experiment with two isolated quantum particles, since we also need the third one to extract some information. The main claim of the paper is to connect the time arrow with such multi-particle processes. We presuppose that some non-associative quantum field theory can exist. The algebra of quantized fields is known only for free fields. The algebra of operators of the strongly interacting quantum fields (in some regimes of quantum chromodynamics, quantum gravity and so on) is much more complicated and can be non-associative. In this regard we have added some text, see the last sentence of introduction and the sentence after (12).
Reviewer 2 Report (Previous Reviewer 1)
Algebraic Entropy and Arrow of Time Referee Report
Brief Summary: In this manuscript, the author considers systems whose wavefunction can be written as a product of at least three wavefunctions such that . Such system may be non-associative. As a result, measurements of results in a number X which is an element of either the complex number, quaternions, or octonions. If one only knows the magnitude of X, N=XX*, then there is an ambiguity associated with the phase angles of X. Placing a maximally mixed distribution on all possible phase vectors captures the uncertainty in X. The relative entropy between passive and active transformations of X gives the algebraic entropy which has at least 18.6 bits of uncertainty for octonions and zero uncertainty for complex numbers and quaternions. This is the unavoidable uncertainty of non-associativity suggested by the author that is related to the 2nd law of thermodynamics.
General Concept Comments:
1. In the paper, the author states “Measurement process means manipulation of a physical system with some device by an observer, i.e. requires at least three operators that are corresponding to these ingredients”. In my previous report, I commented that the basis of irreversibility should be possible without the need of an observer, which put into question the validity of this analysis as a demonstration of the arrow of time. In response, the author points out that observers are not crucial to their analysis. This suggests to me that what is truly important is (1) The system can be described by the product of at least three wavefunctions, (2) the resulting system is non-associative, and (3) X is an element of an octonion. Therefore, the author is not claiming a demonstration of a general arrow of time, but instead, an arrow of time for those systems satisfying conditions (1)-(3).
2. In my previous report, I also pointed out that it is not clear how comparing passive and active transformations of X is related to information that is accessible from continuous measurement. I pointed out my familiarity with process of recovering phase information from measurement by giving phase lift and Kramer-Kronig methods as examples. Unfortunately, the author mistook the mention of these techniques as items that should be addressed instead of elucidating how active and passive transformations are related to information accessible by measurement. Because of this, my previous concern was not addressed. That being said, I still do not understand the relevance of this analysis. Maybe it should be obvious, but it is not clear to me. From the perspective of presentation, I think it’s important that this is made clear. Especially since the calculation giving the main result is very simple. This makes presentation significantly more important, and in its current state, it is unclear why this analysis is interesting.
Author Response
Brief Summary:
In this manuscript, the author considers systems whose wavefunction can be written as a product of at least three wavefunctions. Such system may be non-associative. As a result, measurements of results in a number X which is an element of either the complex number, quaternions, or octonions. If one only knows the magnitude of X, N=XX*, then there is an ambiguity associated with the phase angles of X. Placing a maximally mixed distribution on all possible phase vectors captures the uncertainty in X. The relative entropy between passive and active transformations of X gives the algebraic entropy which has at least 18.6 bits of uncertainty for octonions and zero uncertainty for complex numbers and quaternions. This is the unavoidable uncertainty of non-associativity suggested by the author that is related to the 2nd law of thermodynamics.
General Concept Comments:
- In the paper, the author states “Measurement process means manipulation of a physical system with some device by an observer, i.e. requires at least three operators that are corresponding to these ingredients”. In my previous report, I commented that the basis of irreversibility should be possible without the need of an observer, which put into question the validity of this analysis as a demonstration of the arrow of time. In response, the author points out that observers are not crucial to their analysis. This suggests to me that what is truly important is
(1) The system can be described by the product of at least three wavefunctions,
(2) the resulting system is non-associative, and
(3) X is an element of an octonion.
Therefore, the author is not claiming a demonstration of a general arrow of time, but instead, an arrow of time for those systems satisfying conditions (1)-(3).
The main claim of the paper is to connect the time arrow with multi-particle processes. We presuppose that some non-associative quantum field theory can exist. The algebra of quantized fields is known only for free fields. The algebra of operators of the strongly interacting quantum fields (in some regimes of quantum chromodynamics, quantum gravity and so on) is much more complicated and can be non-associative. In this regard we have added some text, see the last sentence of introduction and the sentence after (12).
2. In my previous report, I also pointed out that it is not clear how comparing passive and active transformations of X is related to information that is accessible from continuous measurement. I pointed out my familiarity with process of recovering phase information from measurement by giving phase lift and Kramer-Kronig methods as examples. Unfortunately, the author mistook the mention of these techniques as items that should be addressed instead of elucidating how active and passive transformations are related to information accessible by measurement. Because of this, my previous concern was not addressed. That being said, I still do not understand the relevance of this analysis. Maybe it should be obvious, but it is not clear to me. From the perspective of presentation, I think it’s important that this is made clear. Especially since the calculation giving the main result is very simple. This makes presentation significantly more important, and in its current state, it is unclear why this analysis is interesting.
The number of generators of coordinate transformations that preserve norms (passive transformations), i.e. the number of independent angles in (9), equals to n(n − 1 )/2 (Formula (16)). So, one can restore all components of a multi-dimensional number if its norm and all SO(n) rotation angles are known. Recall that n = 1, 3 and 7 for complex numbers, quaternions and octonions, respectively.
While for complex numbers and quaternions the number of angles of physically realizable rotations (active rotations, defined by two-side products (7)) are equal to number of angles of accompanied rotations of coordinate axis’s (passive rotations), for octonions generators of automorphism group (active transformations) are less then generators of rotation group (passive transformations). This means that one cannot extract values of all components of an octonionic signal knowing its norm and all active rotation angles.
Reviewer 3 Report (New Reviewer)
Please, see the attached comments.

Author Response
General comments:
The author proposes a way to justify arrows of time via the non-associativity of suitable algebras by using entropic concepts. The paper is rather interesting, and it is certainly suitable for Entropy, in my view. However, prior to its publication, I suggest the author taking into consideration the following specific comments. More importantly, I would strongly suggest partitioning the paper into an Introduction section, and Algebraic Entropy and Arrow of Time section, and a Conclusions section. This would make the paper more readable. Also, I would revise the Abstract by considering my comment 12.
The paper was rewritten according to the suggestion of the referee, thanks.
Specific comments
- Page 1, lines 35-36: Please, rewrite the sentence in a clear manner.
“For our consideration it is not crucial introduction of the observer depended irreversibility, non-associativity could be manifested in any complex system containing more than two particles.” à
See the last sentence of the Introduction was changed.
- Page 1: The authors should state in a very clear manner that their definition of algebraic entropy does not coincide with the one defined by Bellon-Viollet from the very beginning of the manuscript. Furthermore, they should define in a very clear manner what they mean with algebraic entropy at the very beginning. This is a very important in this reader’s opinion.
According to the suggestion of the referee, to avoid confusions, through the text ‘algebraic entropy’ everywhere was replaced by ‘algebraical entropy’.
- Page 2, line 43: I would suggest using “algebraic” everywhere are replace “algebraical”.
Done
- Page 2, line 65:….besides….→…in addition to….
Done
- Page 3, line 71:…intervals….→….interval…
Done
- Page 3, line 72:…[0-2π]….→…[0, 2π]…
- Page 3, nearby Eq. (13): When mentioning the conditional differential entropy, I would suggest adding a suitable general reference like the book by Cover and Thomas.
Done
- Page 4: …we need some measure of information, which is not relative to the coordinate system inuse…. →: …we need some measure of information that is not dependent on the used coordinate system….→
Done
- Page 4, Eq. (16): Remove the comma and insert a period, please.
Done
- Page 4, line 81:…recall…→…Recall….
Done
- Page 4: Given the importance of the groups G2 and SO (7), I would suggest to define these two groups in an explicit manner. If the author feels it would break the flow of the reading, also an Appendix would suffice.
Since for our model the dimensions of the groups G2 and SO(7) (which are well studied in the literature, we mention [40]) is important, we decided not to describe their structure explicitly in this paper. However, we add some more descriptions in the paragraph before (17).
- I would modify a bit the abstract so that 18.6 bit can be understood as explained in Eq. (19). A way to provide a change is by mentioning these groups G2 and SO (7) in the abstract…
Done
Round 2
Reviewer 1 Report (Previous Reviewer 2)
I thank the author for the revisions in the manuscript in response to my comments.
Author Response
Thank you for the review that helped me to improve the paper.
Reviewer 2 Report (Previous Reviewer 1)
Algebraical Entropy and Arrow of Time
General Concept Comments:
-
In my last two reports, I pointed out that the relationship between measurement and active and passive transformations is too vague. The first time, the author misunderstood my concerns so the issue wasn’t addressed. The second time, the author responded with a statement about the number of generators associated with these transformations that essentially contains the same information in the paper. Given this response, I still do not find the presentation sufficient to substantiate the claim that an unavoidable 18.6 bits of entropy exists for any measurement process involving numbers described by octonion algebra. In the following, I will do my best to be more detailed with my concerns and explain why I fundamentally do not recommend this paper for publication.
-
To begin, I want to start with the use of relative entropy as the measure of inaccessible information. To do this, I want to outline the reasoning used in the paper that justifies its use.
a. The only information we have is the magnitude of X (which is an octonion). Therefore, the phase vector is totally unknown thus resulting in Eq.(11) with d=21.b. In principle, one would use the continuous Shannon entropy (?_sh) on Eq.(11) to quantify one’s ignorance about the phase vector.
c. Because ?_sh is not coordinate independent, they turn toward the relative entropy.My problem with this reasoning is the Shannon entropy and relative entropy do not measure the same thing. Given a random variable X with possible outcomes {?_i} one can define probability distributions {?_i} and {?_i} over {?_i}. If one is making predictions about X over many trials using {?_i} and the outcomes are indeed determined by {?_i}, then the Shannon entropy measures how well one can predict the outcome for individual trials using {?_i}. The same is true for any probability distribution defined over {?_i }.
In the above scenario, there’s an inherent uncertainty based on what the probability distribution is. The relative entropy measures the additional uncertainty for any trial given that the true distribution is {?_i} but one is using {?_i} to make predictions. In this case, the relative entropy is written as D(p||q). This is important, because it is not symmetric. Meaning simply swapping to D(q||p) does not mean the same thing. So already, I am confused as to how we went from b. to c. since it seems the only motivation was getting something coordinate independent as opposed to a physical explanation of measurements in a lab to recover missing information.
-
This brings me to my second concern about the use of relative entropy. I will use the notation SO(7) to represent Eq.(11) with d=21 and g2 to represent Eq.(11) with d=14. Given this, SO(7) is associated with passive transformations and g2 is associated with active transformations per the paper. In the original paper, the relative entropy that resulted in 18.6 bits of algebraical entropy was written as D(SO(7)||g2). This implies, if I
understand the authors explanation of active/passive transformations and measurements, that the true phase vector of X belongs to the space of SO(7) but one only has access to the phase vectors associated with the space of g2. Therefore, there is inaccessible information hidden in SO(7) that can’t be reached by g2.
When D(SO(7)||g2) was used, I pointed out in my first report that there is a missing negative sign in the 18.6 bits results. As a response, the author changed the relative entropy to D(g2||SO(7)) to get the correct sign. But now the logic is reversed. It seems as though the true phase vector belongs to g2 and one is using SO(7), so I don’t see how the reasoning of inaccessibility holds. I also find it suspicious that one can just swap SO(7) and g2 without fundamentally changing the reasoning of the calculation.
-
Given my comments in 2. and 3., I also still don’t see how maximally mixed distributions over passive and active transformations result in a measurement process in a lab. Is it the case that there exists some measurement process for which the statistics are determined by the true phase data of X and one only has access to measurements associated with active transformations? And if so, how do maximally mixed distributions play a role in this process? I have quite a bit of exposure to the literature on measuring phase information from physical systems and the explanation here seems opaque to me. I admit I have no familiarity with the concept of Cartan’s smallest exceptional Lie group and that all of my confusion may be a result of that. But this is what I wanted the author to elucidate. If this doesn’t make sense to me, I have a hard time imagining it will be of much use to a broader community beyond those who are familiar. This was another of my main objections previously that I do not think has been addressed. And as I said before, the actual calculation is very simple so I place a high burden on making the presentation accessible.
-
Another concern I have regarding measurement is what if the actual phase vector of X is one that is accessible from active transformations? In that case, wouldn’t it be possible to fully recover the phase information? Why is the 18.6 bits unavoidable? Shouldn’t it actually depend on the state of the physical system? Again, this question may be due to my ignorance, but the claim that any measurement associate with octonions must have 18.6 bits of uncertainty seems too assertive given the present analysis. Overall, I don’t think the evidence presented justifies the claim.
-
I also still have issue with the sentence “in any measurement the three systems: an object, memory (or apparatus) and observer are involved”. This is because my original objection was that irreversibility should not require the existence of an observer. As a response, the author added lines like “Note that for our consideration it is not crucial to introduce the observer dependent irreversibility. Non-associativity could be manifested in any complicated or strongly interacting system containing more than two particles (in some regimes of quantum chromodynamics, quantum gravity and so on).” I appreciate the author specifying more explicitly where this analysis is applicable, but, as I stated in my previous report, the main criteria are (1) The system can be described by the
product of at least three wavefunctions, (2) the resulting system is non-associative, and (3) X is an element of an octonion. I only mention this because (1)-(3) and the second quote above are what really matter. The first quote seems to still function as justification, but it is not necessary because observers aren’t that important overall. This concern is minor compared to those presented in comments (1)-(5).
Overall, I like the idea of non-associativity in strongly interacting systems resulting in an unavoidable production of entropy. I think its an interesting line of thinking, but I don’t think the current analysis justifies the claim made within. For that reason, I do not recommend this paper for publication.
Author Response
I want to thank the referee for his reviews that helped me to improve the paper. Below are his 6 comments with my answers (in bold).
- In my last two reports, I pointed out that the relationship between measurement and active and passive transformations is too vague. The first time, the author misunderstood my concerns so the issue wasn’t addressed. The second time, the author responded with a statement about the number of generators associated with these transformations that essentially contains the same information in the paper. Given this response, I still do not find the presentation sufficient to substantiate the claim that an unavoidable 18.6 bits of entropy exists for any measurement process involving numbers described by octonion algebra. In the following, I will do my best to be more detailed with my concerns and explain why I fundamentally do not recommend this paper for publication.
To clarify our approach, consider as an example a stick with the known length tied up at the origin of 3D Euclidean space. To have information about the location of its second end we need three angles, parameters of the SO(3) group of 3D rotations. Probability that the stick has a specific direction in 3D space can be evaluated using (10) with d = 3. But this uncertainty is not fundamental and we do not need to introduce entropy, since the SO(3) angles can be measured by rotations of our coordinate system (passive transformations) with respect to the coordinate system where the stick is at rest.
Suppose now that due to some mechanism we are trapped in 2D and are able to perform only SO(2) (subgroup of SO(3)) rotations in a plane (active transformations). Having the information that the stick is 3-dimensional we are left with fundamental uncertainty about its actual direction in 3D. This unavoidable uncertainty cannot be characterized by (10), we need the relative entropy (19), obtaining for this case the value (3-1) 2.7 bit = 5.4 bit.
- To begin, I want to start with the use of relative entropy as the measure of inaccessible information. To do this, I want to outline the reasoning used in the paper that justifies its use.
- The only information we have is the magnitude of X (which is an octonion). Therefore, the phase vector is totally unknown thus resulting in Eq. (11) with d=21.
- In principle, one would use the continuous Shannon entropy (?_sh) on Eq. (11) to quantify one’s ignorance about the phase vector.
- Because ?_sh is not coordinate independent, they turn toward the relative entropy.
My problem with this reasoning is the Shannon entropy and relative entropy do not measure the same thing. Given a random variable X with possible outcomes {?_i} one can define probability distributions {?_i} and {?_i} over {?_i}. If one is making predictions about X over many trials using {?_i} and the outcomes are indeed determined by {?_i}, then the Shannon entropy measures how well one can predict the outcome for individual trials using {?_i}. The same is true for any probability distribution defined over {?_i}.
In the above scenario, there’s an inherent uncertainty based on what the probability distribution is. The relative entropy measures the additional uncertainty for any trial given that the true distribution is {?_i} but one is using {?_i} to make predictions. In this case, the relative entropy is written as D(p||q). This is important, because it is not symmetric. Meaning simply swapping to D(q||p) does not mean the same thing. So already, I am confused as to how we went from b. to c. since it seems the only motivation was getting something coordinate independent as opposed to a physical explanation of measurements in a lab to recover missing information.
As it was explained in our example above, to define fundamental algebraical uncertainty we need to introduce the measure of information that cannot be retrieved by passive transformations.
- This brings me to my second concern about the use of relative entropy. I will use the notation SO(7) to represent Eq.(11) with d=21 and g2 to represent Eq.(11) with d=14. Given this, SO(7) is associated with passive transformations and g2 is associated with active transformations per the paper. In the original paper, the relative entropy that resulted in 18.6 bits of algebraical entropy was written as D(SO(7)||g2). This implies, if I understand the authors explanation of active/passive transformations and measurements, that the true phase vector of X belongs to the space of SO(7) but one only has access to the phase vectors associated with the space of g2. Therefore, there is inaccessible information hidden in SO(7) that can’t be reached by g2.
When D(SO(7)||g2) was used, I pointed out in my first report that there is a missing negative sign in the 18.6 bits results. As a response, the author changed the relative entropy to D(g2||SO(7)) to get the correct sign. But now the logic is reversed. It seems as though the true phase vector belongs to g2 and one is using SO(7), so I don’t see how the reasoning of inaccessibility holds. I also find it suspicious that one can just swap SO(7) and g2 without fundamentally changing the reasoning of the calculation.
The true phase vector of X belongs to the space of SO(7) but one only has access to the phase vectors associated with the space of g2. Therefore, there is inaccessible information hidden in SO(7) that can’t be reached by g2.
For the case of our example, we know that the stick is 3-dimensional (7-sphere for octonions), but can really measure its components only in 2D (g2 space in our paper). Then we need to compare distributions of SO(2) angles with SO(3) ones (i.e. D(g2||SO(7)) for octonions).
- Given my comments in 2. and 3., I also still don’t see how maximally mixed distributions over passive and active transformations result in a measurement process in a lab. Is it the case that there exists some measurement process for which the statistics are determined by the true phase data of X and one only has access to measurements associated with active transformations? And if so, how do maximally mixed distributions play a role in this process? I have quite a bit of exposure to the literature on measuring phase information from physical systems and the explanation here seems opaque to me. I admit I have no familiarity with the concept of Cartan’s smallest exceptional Lie group and that all of my confusion may be a result of that. But this is what I wanted the author to elucidate. If this doesn’t make sense to me, I have a hard time imagining it will be of much use to a broader community beyond those who are familiar. This was another of my main objections previously that I do not think has been addressed. And as I said before, the actual calculation is very simple so I place a high burden on making the presentation accessible.
Obtained results are general and do not depend on the specific properties of the groups of active and passive transformations, SO(2) and SO(3) in the stick example above. So, in my opinion we don’t need to insert separate reviews of SO(7) and g2 groups in our short paper.
- Another concern I have regarding measurement is what if the actual phase vector of X is one that is accessible from active transformations? In that case, wouldn’t it be possible to fully recover the phase information? Why is the 18.6 bits unavoidable? Shouldn’t it actually depend on the state of the physical system? Again, this question may be due to my ignorance, but the claim that any measurement associate with octonions must have 18.6 bits of uncertainty seems too assertive given the present analysis. Overall, I don’t think the evidence presented justifies the claim.
Let us translate this question for my example of the 3D stick that is observed by 2D creatures: What if the stick exactly lies in the observed plane? This accidental coincidence does not mean that entropy corresponding to 3D rotations disappears.
- I also still have issue with the sentence “in any measurement the three systems: an object, memory (or apparatus) and observer are involved”. This is because my original objection was that irreversibility should not require the existence of an observer. As a response, the author added lines like “Note that for our consideration it is not crucial to introduce the observer dependent irreversibility. Non-associativity could be manifested in any complicated or strongly interacting system containing more than two particles (in some regimes of quantum chromodynamics, quantum gravity and so on).” I appreciate the author specifying more explicitly where this analysis is applicable, but, as I stated in my previous report, the main criteria are (1) The system can be described by the product of at least three wavefunctions, (2) the resulting system is non-associative, and (3) X is an element of an octonion. I only mention this because (1)-(3) and the second quote above are what really matter. The first quote seems to still function as justification, but it is not necessary because observers aren’t that important overall. This concern is minor compared to those presented in comments (1)-(5).
We agree that this issue, mentioned by the referee, is not clearly described in the paper. In revised version the last paragraph of the introduction was rewritten in the form:
“In this paper we consider the possibility to link the microscopic entropy and the arrow of time with the non-associativity of nature at a fundamental level. Measurement process means manipulation of a physical system with some device by an observer, i.e. requires at least three operators that are corresponding to these ingredients. Final result of a measurement is some real number obtained from multi-component functions (e.g. the norm of a wave function), i.e. any physical observable depends also on the algebra we use.
Note that for our consideration it is not crucial to introduce the observer dependent irreversibility, non-associativity could be manifested in any complicated or strongly interacted system, containing more than two particles (e.g., in some regimes of quantum chromodynamics, quantum gravity and so on). Any particle (an excitation of quantum field), before a measurement by interactions with another particle, can be considered as entangled with the rest of the universe (multi-particle system). So, any measurement should be described by the product of at least three wave functions and we presuppose that some non-associative quantum field theory can exist.”
Reviewer 3 Report (New Reviewer)
The authors have sufficiently improved the paper according to our suggestions.
Author Response
Thank you for the review that helped me to improve the paper.
Round 3
Reviewer 2 Report (Previous Reviewer 1)
Algebraical Entropy and Arrow of Time
General Concept Comments:
1. After reviewing the authors response to my last report, I am still not convinced that this article warrants publication in entropy. In what follows is a summary of why I have taken this position.
2. In my first response, I commented that the description of the measurement process is the justification for this analysis. More specifically, it connects the arrow of time to non-associativity which leads to the justification of considering octonions. Initially, my concern was that the description of measurement as requiring an observer was a flaw since irreversibility should be possible independent of an observer. In response, the author adjusted their position to claim that the true requirements that justify the analysis are (1) The system can be described by the product of at least three wavefunctions, (2) the resulting system is non-associative, and (3) X is an element of an octonion. They also mentioned that “non-associativity could be manifested in any complicated or strongly interacting system containing more than two particles (e.g., in some regimes of quantum chromodynamics, quantum gravity and so on)”, though no references where given.
Ultimately, what I original saw was a weak connection between physics and mathematics. The analysis seemed like an interesting exercise with octonions that largely felt disconnected from physics given the original justification for non-associativity. The subsequent adjustment to conditions (1)-(3) changes the original claim, but it relegates the applicability of the results to what seems like niche circumstances requiring non-associativity. As someone who is familiar with current and historic research on reversibility and the 2nd law, I do not think this analysis will garner enough interest to warrant publication.
3. I also pointed out in my original report that the connection between passive and active transformations with information that can be recovered from continuous measurements in a lab seemed vague and not clearly explained. Given all of the responses from the author thus far, I still do not think my concern has been addressed. After my first report, the concern was not addressed due to miscommunication. In the second response from the author, I was essentially given the same information in the paper that, I felt, did not give any additional information to address my concern. In the last response, the author gave an explicit example which I will now discuss to demonstrate that my concern has still not been addressed.
In the example, the author uses a stick in R^3. They say one end of the stick is located at the origin of some coordinate system while the other end is located somewhere in the space. They then say “we do need to introduce entropy since the SO(3) angles (determining the location of the other end of the stick) can be measured by rotations of our coordinated system (passive transformations) with respect to the coordinate system where the stick is at rest”. Given this description, I have earnestly tried to discern what it means to measure the location of the other end of the stick using passive transformations. Are they saying something like the stick’s end has a location in R^3 and there exists two coordinates systems that are unitarily related to each other? The first is the fixed system where the stick is at rest (say in the negative z direction). The second coordinate system is accessible to the experimenter and begins in some orientation w.r.t to the first, and the choice of the initial orientation fixes the global phase? One then rotates the second coordinate system using SO(3) symmetries until say the z-direction points in the same direction as the end of the stick thus finding the phase vector w.r.t to the choice of global phase? Or, are the passive transformations associated with the orientation of a measurement device for which outcomes depend on the orientation so that a phase vector is established given measurement outcomes? I honestly don’t know the answer which points to the vagueness that I think needs to be clearly explained.
In addition to my confusion on how phase is measured from passive transformations, the author’s example also provides an explanation for the relation between passive and active transformations. They say “suppose now that due to some mechanism we are trapped in 2D and are able to perform only SO(2) (subgroup of SO(3)) rotations in a plane (active transformations)”. Now I am very confused because the restriction to 2D has introduced active transformations which tells us about our measurement process to determine the location of the stick. But what is the connection between measuring the stick’s location with full access to R^3 using passive transformations to now being in 2D and using active transformations? Now my attempts to understand measuring phase using passive transformations in the previous example don’t seem to apply since accessibility is determined by active transformations. Ultimately, the response from the author has only left me more confused.
4. Since I can’t establish a clear understanding between passive/active transformations and measurement, I don’t see how the relative entropy quantifies an unavoidable entropy production. In my last report, I brought up the fact that the justification for the use of relative entropy was also not clear. This was my response 2. The author replied by saying that this concern was addressed by the example, but since the example did not clarify things for me, the comment was not addressed.
For the reasons given above and in my previous reports, I do not think my concerns have been addressed. For this reason, I do not feel confident that the analysis given justifies the claim of an unavoidable 18.6 bits of entropy being produced in the circumstances of non-associativity that leads to octonions. As I stated in my first report, this seems like an interesting mathematical exercise with loose connections to physics.
This manuscript is a resubmission of an earlier submission. The following is a list of the peer review reports and author responses from that submission.
Round 1
Reviewer 1 Report
Algebraic Entropy and Arrow of Time Referee Report
Brief Summary: In this manuscript, the author suggests an algebraic entropy as an explanation for the arrow of time as it relates to the 2nd law of thermodynamics. The main idea is that any measurement requires at least three parties: an object being observed , a measurement device , and an observer . Because of this, there is an inherent uncertainty for any measurement process caused by the ambiguity of non-associativity of the product . For any measurement of , one receives a number X which is an element of either the complex numbers, quaternions, or octonions. If one only knows the magnitude of X, N=XX*, then there is an ambiguity associated with the phase angles of X. Placing a maximally mixed distribution on all possible phase vectors captures the uncertainty in X. The relative entropy between passive and active transformations of X gives the algebraic entropy which has at least 18.6 bits of uncertainty for octonions and zero uncertainty for complex numbers and quaternions. This is the unavoidable uncertainty of non-associativity suggested by the author that is related to the 2nd law of thermodynamics.
General Concept Comments:
1. One major criticism is the claim that any measurement process requires at least three parties: observer, measurement device, and object, and that this requirement is the basis for irreversibility. This seems to me to imply that irreversibility is not possible without an observer for example. Also, I’m not aware of such a requirement for measurement processes in the literature. This is a major weakness because this description of a measurement process establishes the connection between non-associativity and the arrow of time, and it justifies the use of the octonion algebraic structure.
The disconnect between the claim of reversibility and the main analysis of the paper seems to further show itself in the distribution over phases first given by Eq. (11). Here, it is clear that the uncertainty is independent of the magnitude N. As such, it implies that the uncertainty is exclusively a consequence of X, which is an octonion (since it is the only normed algebra that gives a non-zero value of Eq.(14).). This means that any measurement that results in an octonion will have this uncertainty independent of the specific measurement process. This is the basis of the claim that the 18.6 bits of uncertainty is unavoidable. Because of this, it is crucial that connection between measurement and octonion/non-associativity is clearly justified.
2. Given comment 1., the analysis on page four of the manuscript may be interesting as a separate investigation of octonion structure independent of the arrow of time. It is unclear however how the comparison of active and passive transformations of X is related to information that is accessible from continuous measurement. I am familiar with literature related to recovering phase information from amplitude data like the Kramer’s-Kronig relations or phase lift techniques, but I’m not sure how this is related to that. Some further explanation by the author might be helpful.
Specific comments:
1. I think there should be a sum over n in Eqs. (7) and (15)
2. I think there’s a missing negative sign in Eq. (19)
Recommendation: Because of the issues discussed in comment (1), I do not recommend that this paper be published in Entropy. I think the author falls short in their central claim that they have shown an unavoidable 18.6 bits of entropy in any measurement process. Mainly because the connection between non-associativity/octonions and measurement seems weak. More effort in justifying this connection is needed.
Reviewer 2 Report
The manuscript attempts to relate the entropy generated in the quantum measurement process to the algebra of octonions, and derives a a very particular example for the amount of entropy generated, as a KL divergence. The manuscript contains a helpful review of the algebra of octonions, in light of the algebra of C numbers as well as quartenions.
While the review of the algebra used is indeed helpful, it appears not very clear how the objectives of the study and conclusions drawn are related to the particular example presented. Primarily, the manuscript lacks details of how the quantum measurement process (involving a system, apparatus and the observer) can be related to the algebra of octonions. The comparison in terms of KL divergence made between two probability distributions are also not sufficiently justified in terms of the quantum measurement process. As a consequence, the particular conclusions drawn appears to be far fetched from the results of the calculations. Therefore I believe the manuscript is not publishable in its present form.